# Downregulation of MMP-9 Enhances the Anti-Migratory Effect of Cyclophosphamide in MDA-MB-231 and MCF-7 Breast Cancer Cell Lines

**DOI:** 10.3390/ijms222312783

**Published:** 2021-11-26

**Authors:** Magdalena Izdebska, Wioletta Zielińska, Adrian Krajewski, Marta Hałas-Wiśniewska, Klaudia Mikołajczyk, Maciej Gagat, Alina Grzanka

**Affiliations:** Department of Histology and Embryology, Faculty of Medicine, Collegium Medicum in Bydgoszcz, Nicolaus Copernicus University in Toruń, 87-100 Toruń, Poland; mizdebska@cm.umk.pl (M.I.); w.zielinska@cm.umk.pl (W.Z.); adrian.krajewski@cm.umk.pl (A.K.); mhalas@cm.umk.pl (M.H.-W.); klaudia.mikolajczyk@cm.umk.pl (K.M.); agrzanka@cm.umk.pl (A.G.)

**Keywords:** MMP-9, CP, CK19, breast cancer, metastasis

## Abstract

Metastasis is one of the most urgent issues in breast cancer patients. One of the factors necessary in the migration process is the remodeling of the extracellular matrix (ECM). Metalloproteinases (MMPs) can break down the elements of the ECM, which facilitates cell movement. Many highly aggressive tumors are characterized by high levels of MMPs. In the case of breast cancer, the association between MMP-9 and the migration potential and invasiveness of cells has been demonstrated. In addition, reports indicating increased migration of breast cancer cells after the administration of the commonly used cytostatic cyclophosphamide (CP) are particularly disturbing. Hence, our research aimed to assess the effect of CP treatment on MDA-MB-231 and MCF-7 cells and how this response is influenced by the downregulation of the MMP-9 level. The obtained results suggest that CP causes a decrease in the survival of breast cancer cells of various invasiveness, and the downregulation of MMP-9 enhances this effect, mainly by inducing apoptosis. Moreover, in the group of MMP-9 siRNA-transfected CP-treated cells, a more severe reduction in invasion and migration of cells of both lines was observed, as indicated by the migration and invasion transwell assays and Wound healing assay. Hence, we suggest that CP alone may not result in satisfactory therapeutic effects. On the other hand, the use of combination therapy targeting MMP-9, together with the CP, could improve the effectiveness of the treatment. Additionally, we confirmed a relationship between the levels of MMP-9 and cytokeratin 19 (CK19).

## 1. Introduction

One of the most important problems of cancer patients is metastasis. It contributes to almost 66.7% of cancer mortality [1]. It also applies to breast cancers tumors, 81% of which are invasive. Although in patients with lesions limited only to the primary site the 5-year survival is relatively high, in the case of metastases it drops to only 30%. Thus, the search for new therapy goals based on broadly understood cancer cell biology, with particular emphasis on molecular factors involved in the emergence of secondary tumor outbreaks is reasonable. Highly invasive cancer cells can change the phenotype from epithelial to mesenchymal. These changes are called the epithelial-mesenchymal transition (EMT), during which cells acquire motor skills. The formation of metastases is a cascade of closely consecutive molecular and morphological changes, including the reconstruction of the cytoskeleton and extracellular matrix (ECM) [2,3]. Metalloproteinases (MMPs) are zinc-dependent endopeptidases responsible for physiological and pathophysiological tissue remodeling [4]. According to available literature reports, breast cancer cells of various invasiveness are characterized by variable expression of MMP, e.g., MMP-1, MMP-2, MMP-9, MMP-7, MMP-10, and MMP-19 [5]. Individual MMP isoenzymes have different substrate specificity, e.g., MMP-1 shows hydrolytic activity against collagens (collagenases). During metastasis, MMPs degrade ECM proteins. It opens pathways for cancer cells that move using actin-rich protrusions, e.g., filopodia [6,7]. Changes in MMPs levels have been observed in many cancers, such as ovarian and colorectal cancer [8]. MMPs enzymatic activity also correlates with the stage of cancer, increased invasiveness, and lower survival rate. These factors emphasize the value of MMPs in the prognosis and monitoring of the effectiveness of cancer treatment [9,10]. MMP-1 and MMP-9 are also very important in these processes, especially in cancers with different receptor profiles, e.g., progesterone positive (ER+) and triple-negative breast cancer (TNBC) [11,12]. MMP-9 is crucial in metastasis due to the high potential for ECM degradation (various substrates: gelatin types I and V, collagen types IV and V, fibronectin). Moreover, its high expression correlates with a worse prognosis among patients with breast, kidney, lung, and ovarian cancers [9]. The importance of cytokeratins (CK) is also of great interest, both in the context of the migration process and their prognostic value. The relationship between MMPs and CKs raises many questions regarding the nature and function of this interaction. The common point was indicated in the research on sentinel lymph nodes specimens obtained from breast cancer patients. The study presents the prognostic value of MMP-9 but also its correlation with CK19 [13].

Our study aimed to investigate how changes in the expression level and enzymatic activity of MMPs affect the proliferative and migratory capacity and thus metastatic potential of breast cancer cells. Additionally, we treated cells with Cyclophosphamide (CP) used in chemotherapy of various types of cancer, including breast cancer. Recent reports indicate high toxicity of CP and increased migration of the CP-treated breast cancer cells [14,15]. Therefore, we wanted to investigate how reduced MMP-9 expression affects the response of breast cancer cells (MDA-MB-231 and MCF-7) to CP treatment and whether there is a correlation between MMP-9 and CK19.

## 2. Results

### 2.1. Breast Cancers Are Characterized by Increased Levels of MMP-9

The first step of the study was the transfection of MCF-10A, MCF-7, and MDA-MB-231 cells with the MMP-9 siRNA and control siRNA. The obtained results were compared with non-transfected cells. Densitometric analysis of the bands reflecting protein levels obtained by Western blot indicated a decrease in MMP-9 level in cells of the MCF-7 and MDA-MB-231 lines transfected with MMP-9 siRNA compared to cells transfected with control siRNA and non-transfected cells. We did not observe any significant differences between control siRNA-transfected and non-transfected cells of neither MCF-7 nor MDA-MB-231 cells. Thus, in further analyzes (except MTT assay) non-transfected cells were used as a control (Figure 1).

Research indicates that the expression of MMP-9 in normal breast tissue is relatively low [16,17]. Similarly, we observed that non-tumorigenic cells of the MCF-10A line are characterized by the low levels of the metalloproteinase (Figure 1A). In turn, MMP-9 expression is significantly increased in breast cancer cells. In addition, the study showed that MMP-9 level varies in different molecular subtypes of breast cancer. Significant MMP-9 overexpression characterizes the triple-negative MDA-MB-231 breast cancer cell line (Figure 1C) [16,17]. In MCF-7, a model of the estrogen and progesterone receptor positive slightly aggressive and non-invasive cell line [18], the expression of MMP-9, was also elevated compared to MCF-10A cells (Figure 1B). In non-transfected cancer cells, we also observed that CP did not reduce the level of MMP-9. In the MCF-7 cell line, it was even slightly elevated (without statistical significance) (Figure 1A–C).

In addition, analysis of the RNA-seq data confirmed the obtained results and showed that MMP-9 is overexpressed in the TCGA breast cancer cohort compared to normal tissue (Figure 2A). At the same time, we did not find differences in MMP-9 mRNA expression depending on the tumor stage (Figure 2B). However, the differentiation of PAM50 showed that MMP-9 expression is significantly higher in basal breast cancer compared to the luminal A subtype (Figure 2C). Then, we checked if the overexpression of MMP-9 has an impact on breast cancer patients’ overall survival. The analysis revealed no significant difference between OS and MMP-9 mRNA expression (Figure 2D).

To compare mRNA data with protein expression we analyzed CPTAC data using the UALCAN database. We revealed that the expression of MMP-9 is significantly lower among breast cancer cohorts compared to normal tissue. Stage differentiation confirmed the results of TCGA analysis, and no significant differences were revealed between particular tumor stages. Additionally, only the luminal subtypes showed a significant MMP-9 downregulation (Figure 2E–G).

### 2.2. Downregulation of MMP-9 and Treatment with CP Reduce Cell Viability and Colony Formation, Induce Apoptosis, but Do Not Change the Distribution of Cell Cycle Phases

Parallel to the transfection, we determined the dose of CP inhibiting cell survival by approximately 50% (IC50). For this purpose, we used the MTT assay. A 5 mM dose of the cytostatic in transfected cells reduces the cell survival to about 50%. As shown in Figure 3A, the survival of transfected cells of the MCF-7 line after treatment with 5 mM CP was 55.77 ± 10.47%, while in MDA-MB-231, it reached 58.29 ± 13.57% (Figure 3A). Based on the MTT assay results and calculated IC50, the 5 mM dose of CP was selected for further experiments.

In the next step, we checked whether the downregulation of MMP-9 influences the cell death of MCF-7 and MDA-MB-231 cells and affects their response to 5 mM CP. The standard AV/PI double-labeling method was used to determine the percentage of live, apoptotic and necrotic cells. In the population of MCF-7 cells, a statistically significant decrease in the percentage of live cells was found in the MMP-9 siRNA transfected and treated with 5 mM CP group (58.57 ± 18.47%). It was accompanied by a statistically significant increase in the apoptotic cell population (43.60 ± 11.68%). Moreover, in non-transfected MCF-7 cells, a CP-induced gain in the percentage of necrotic cells (9.97 ± 2.69%) was observed. Similarly, CP caused a statistically significant decrease in the population of living cells in non-transfected, and MMP-9 siRNA transfected MDA-MB-231 cells in comparison to untreated cells (79.50 ± 5.35 vs. 95.01 ± 1.51 and 71.15 ± 9.48 vs. 91.38 ± 3.94). The decrease in the percentage of viable cells was also associated with an increase in the population of apoptotic cells in both, non-transfected (21.68 ± 2.66% vs. 4.89 ± 0.78%) and transfected (31.53 ± 9.41% vs. 8.24 ± 3.72%) groups compared to the control (Figure 3B). However, the observed changes were not accompanied by any statistically significant differences in the distribution of the cell cycle phases (Figure 3C). Representative plots were presented in the Appendix A. The colony formation test showed that CP treatment in non-transfected MCF-7 and MDA-MB-231 cells decreased the relative number of colonies formed. The reduction was even greater in cells with downregulated MMP-9 level (Figure 3D).

### 2.3. MMP-9 Downregulation Inhibits Migration and Invasion of MCF-7 and MDA-MB-231 Cells

It was previously reported that high levels of MMP-9 promote the aggressiveness of breast cancer [9]. We decided to test whether the downregulation of MMP-9 affects the ability of MCF-7 and MDA-MB-231 cells to migrate and invade through the basal membrane. The transwell migration and Matrigel invasion assays showed that the reduction of MMP-9 level followed by the treatment with CP significantly decreased the cell migration capacity (Figure 4A) and invasiveness (Figure 4B). The wound healing assay confirmed the results. In both breast cancer cell lines, the use of 5 mM CP prolonged the process of wound closure (Figure 5).

### 2.4. Downregulation of MMP-9 Inhibits the EMT Process in Breast Cancer Cells

In further experiments, we wanted to investigate whether the observed inhibition of the migration was connected with cytoskeleton rearrangement and EMT blockage. Microscopic analysis revealed that MMP-9 silencing resulted in the reorganization of the actin cytoskeleton (Figure 6C). Additionally, Western blot and immunofluorescence staining indicated the reduction in the level of EMT markers such as vimentin (Figure 6A,D) and N-cadherin (Figure 6B,D) in cells after MMP-9 siRNA transfection and CP treatment. There was a statistically significant decrease in the fluorescence intensity of these proteins in both MMP-9 siRNA transfected CP-treated and untreated MCF-7 and MDA-MB-231 cells. However, CP treatment even intensified the observed changes (Figure 6A,B).

### 2.5. Expression of Cytokeratin 19 in Breast Cancer and Its Correlation with MMP-9

Analysis of the RNA-seq data showed that KRT19 (CK19) mRNA is overexpressed in the TCGA breast cancer cohort compared to normal tissue (Figure 7A). As in the case of MMP-9, there were no significant differences in mRNA expression depending on the tumor stage (Figure 7B). We did not find differences in KRT19 (CK19) mRNA expression depending on subtypes of breast cancer (Figure 7C). Additionally, we analyzed whether the overexpression of KRT19 (CK19) has a significant impact on breast cancer patients’ overall survival. Interestingly, patients with higher KRT19 (CK19) expression were characterized by better outcomes (*p* = 0.0068) (Figure 7D). However, the accuracy of the CAPTAC analysis was limited due to the small number of cases available in the database (Figure 7E–G).

Western blot analysis revealed that the level of CK19 in MCF-7 and MDA-MB-231 cells was reduced after the downregulation of MMP-9 with the use of siRNA (Figure 8B). The obtained results were confirmed by cytometric measurements of fluorescence intensity and images from a confocal microscope (Figure 8A).

## 3. Discussion

Inhibition of tumor metastasis is a desirable goal of anti-cancer therapy. It is especially important in triple-negative breast cancer, which is characterized by very high aggressiveness and mortality [19]. Invasive cells can move through the reorganization of the cytoskeleton, which allows the formation of migratory protrusions. MMPs facilitate this process by digesting the extracellular matrix, which paves the way for cancer cells. In vitro studies and analysis of available data on breast cancer patients presented in this paper indicate high levels of MMP-9 in breast tumors [8,9,12]. MMP-9 is classified as a tumor biomarker and prognostic factor, and its regulation may become a therapeutic target [20]. In 2016, Moirangthem et al. showed that knockdown of MMP-9 in the breast cancer cell line MDA-MB-231 may reduce progression by increasing cell-to-cell adhesion and regulating EMT markers, including urokinase receptor (uPAR), E-cadherin, vimentin, and Snail. In addition, an enhanced anti-metastatic effect after a simultaneous knockdown of the plasminogen activator urokinase (uPA) and MMP-9 was indicated [21].

Under unfavorable conditions such as hypoxia or inflammation, intracellular signaling pathways trigger the expression of MMP-9 [22]. These include the phosphoinositide 3 (PI3K)/protein kinase B (AKT), mitogen-activated protein kinases (MAPK)/extracellular signal-regulated kinases (ERK), and nuclear factor-kappa B (NF-kB) pathways [22]. Numerous scientific reports indicate that different compounds target PI3K/AKT/NFκB, AKT/mTOR pathways reducing MMP-9 expression. Wu et al. noticed that natural herbal flavonoid—luteolin suppresses triple-negative breast cancer cell proliferation and metastasis by the downregulation of MMP-9 expression via the AKT/mTOR signaling pathway [23]. On the other hand, Zeng et al. indicated the reduction of MMP-9 levels in MDA-MB-231 and MCF-7 cells after treatment with tectorigenin [24]. Furthermore, piceatannol inhibits the invasion of breast cancer cells through the PI3K/AKT and NF-κB pathways and inhibition of MMP-9 [25]. It is in line with our results as MMP-9 downregulation reduced the migration potential of breast cancer cells of different aggressiveness. Moreover, we decided to check whether CP, a cytostatic commonly used in breast cancer, may intensify the changes observed after MMP-9 silencing or abolish it. This aspect was of particular interest as there were some disturbing reports on the pro-cancerous effect of CP [14]. Studies by Hung et al. indicate that CP can induce breast cancer metastasis. They assessed the results of CP treatment on the migration of cancer cells and its correlation with the chemokine receptor 4 (CXCR4), which is considered a biomarker of cancer metastasis [14]. The obtained results indicate that in MDA-MB-231 cells, increasing concentrations of CP induced the expression of CXCR4 and thus facilitated cell migration. Moreover, the expression of MMP-9 in CP-treated cells was elevated. Zhang et al. also observed that CP increases the levels of MMP-9. By examining the effect of CP in the kidney tissue of rats with Diabetic nephropathy, they indicated decreased expression of TGF-β1 and increased level of MMP-9 [26]. Our data also suggest that CP can indeed increase the expression of MMP-9, and its activity in monotherapy does not result in a statistically significant anti-tumor effect. Perhaps the use of combination therapy along with the agent targeting MMP-9 would be more effective.

On the other hand, Guo et al. showed that a newly described chemical synthetic peptide (E5) inhibits the CXCR4/CXCL12 axis in breast cancer both in vitro and in vivo. E5 was able to bind specifically to the 4T1 breast cancer line, inhibit migration by reducing CXCR4 expression. Its combination with paclitaxel or CP significantly inhibited tumor growth in a breast cancer model [27]. Moreover, the preclinical studies indicate the therapeutic effect and low toxicity of the combined chemotherapy with CP and celecoxib in the case of breast tumors. The combination was anti-angiogenic and was well tolerated [28]. CP was also used in combination with Doxorubicin, 5-fluorouracil, Methotrexate, Docetaxel, and Paclitaxel [29,30]. Currently, the application of such therapy is limited due to serious side effects, liver damage in particular. However, the possible combination with other compounds or factors altering the tumor environment makes CP an important cytostatic in anti-cancer therapy. Cancer cell apoptosis and suppression of tumor growth were observed after implantation of nude mice with MDA-MB-231 cells and treatment with a combination of CP with Chalone 19 peptide [31]. In turn, Agrawal et al. found that insulin increases the cytotoxic effect of 5-fluorouracil and CP and enhances the apoptosis in the MCF-7 cell line. We also confirmed pro-apoptotic and antimetastatic effects of a 5 mM CP concentration in MCF-7 and MDA-MB-231 cells with reduced MMP-9 expression. Analysis of the percentage of apoptotic cells showed a statistically significant increase only in the group of cells transfected with MMP-9 siRNA treated with CP. Moreover, the downregulation of MMP-9 followed by the cytostatic treatment reduced migration and invasion (wound healing assay, chemotaxis test) and influenced the level of EMT markers in cells of both breast cancer cell lines. Vimentin is an intermediate filament protein and an established EMT marker. Similarly, the high level of N-cadherin indicates the cell’s pro-migratory phenotype [32]. The observed decrease of the EMT markers in cells transfected with MMP-9 siRNA treated with CP proves the effectiveness of such approach.

What we also found interesting was a decrease in the level of CK19 after the downregulation of MMP-9. TCGA dataset confirmed that breast cancer is characterized by the overexpression of MMP-9 and CK19 mRNA. However, it has no impact on breast patients’ survival. Although we observe the significant difference between low and high KRT19 (CK19) expression survival, the accuracy of these findings is limited due to the large disproportion between-group strength (929 vs. 118). In turn, CPTAC data suggest the downregulation of MMP-9 in breast cancer patients and upregulation of KRT19 (CK19). However, because of the size of the cohort and the lack of other clinical parameters, we cannot provide any conclusive remarks from this analysis. Both MMP-9 and CK19 are considered negative prognostic markers, but the effect of CK19 in breast cancers seems to be more complex and ambiguous. Some studies show the oncogenic functions of CK19 associated with the stabilization of the HER2/ERK signaling pathway [33]. On the other hand, CK19 can bind β-catenin/RAC1 and control NUMB transcriptional activity, a protein that is an upstream Notch pathway inhibitor [34]. Moreover, the silencing of CK19 in breast cancer cells results in Akt/PTEN pathway [35].

Murawski et al. assessed the importance of MMP-9 expression as a supporting marker for CK19 mRNA in sentinel lymph nodes in breast cancer patients and the link between this expression and clinical-pathological data. The results indicate a strong correlation of MMP-9 with CK19 mRNA in breast carcinoma [13]. In turn, Joseph et al. indicate an increased level of CK17, which was associated with the increased expression of MMP-9. Patients with breast tumors characterized by high expression of CK17 and CK5/6 show high mortality, which indicates the correlation between cytokeratins and metalloproteinases [12].

## 4. Materials and Methods

### 4.1. Cell Culture

The human breast cancer MCF-7 (HTB-22) and MDA-MB-231 (HTB-26) cell lines were obtained from American Type Culture Collection (ATCC, Manassas, VA, USA). The control was a non-tumorigenic epithelial MCF-10A (EP-CL-0525) cell line purchased from Elabscience Biotechnology Inc. All cell lines were cultured according to the manufacturer’s instructions. MCF-7 cells were maintained in Eagle’s Minimum Essential Medium (EMEM, cat. No. 10-009-CV, Corning, New York, NY, USA) supplemented with 0.01 mg/mL Recombinant Human Insulin (rHu Insulin, cat. No. HY-P1156/CS-7906, MedChemExpress, Monmouth Junction, NJ, USA), 10% Fetal Bovine Serum (FBS, cat. No. 35-079-CV, Corning, New York, NY, USA) and 1% Penicillin-Streptomycin Solution (P/S, cat. No. 30-002-CI, Corning, New York, NY, USA). The MDA-MB-231 cell line was cultured in Dulbecco’s Modified Eagle Medium (DMEM, cat. No. 12-604F, Lonza, Basel, Switzerland) with 10% FBS and 1% P/S. MCF-10A cell line was grown in monolayer in DMEM/F12 (1:1 mixture of DMEM and Ham’s F12, cat. No. 10-090-CV, Corning, New York, NY, USA) with 5% Horse Serum (HS, cat. No. 16050-122, Thermo Fisher Scientific, Waltham, MA, USA), 20 ng/mL Human Epidermal Growth Factor (EGF, cat. No. AF-100-15-100UG, Peprotech Ec Ltd., London, UK), 0.5 µg/mL Hydrocortisone (cat. No. HY-N0583/CS-2226, MedChemExpress, Monmouth Junction, NJ, USA), 10 µg/mL rHu Insulin (cat. No. HY-P1156/CS-7906, MedChemExpress, Monmouth Junction, NJ, USA), 1% Non-Essential Amino Acid (NEAA, cat. No. M7145-100 mL, Merck KGaA, Darmstadt, Germany) and 1% P/S. The culture was carried out in a 95% humidified atmosphere, 5% CO_2_ at 37 °C. The cells were cultivated up to five passages and regularly tested for mycoplasma contamination using 4′,6-Diamidino-2-phenylindole dihydrochloride-staining solution (DAPI, D9542, Merck KGaA, Darmstadt, Germany). The cells were subcultured after reaching about 80% confluency using Trypsin-Ethylenediamine tetraacetic acid (EDTA, cat. No. 25-053-CI, Corning, New York, NY, USA) and on T-25 cm^2^ flasks (cat. No. 156367, Thermo Fisher Scientific, Waltham, MA, USA) or 24, 12- and 6-well plates (cat. No. SIAL0524, cat. No. SIAL0513, cat. No. SIAL0516, Merck KGaA, Darmstadt, Germany) depending on the assay and in cell type-dependent density. Before the experiments, optimal density was investigated by seeding cells at different concentrations and evaluated visually in light Axio Observer Z1 inverted motorized microscope (Zeiss, Oberkochen, Germany). Twenty-four hours after seeding, the cells were treated with different concentrations of CP (cat. No. T0707, TargetMol, Wellesley Hills, MA, USA) selected based on the available literature reports including declared CP concentrations in plasma and the Thiazol-2-yl-2,5-diphenyl tetrazolium bromide (MTT) assay and incubated for the next twenty-four hours. Control cell’s populations were cultured in the same conditions and time without the addition of CP.

### 4.2. Cell Transfection

The cells were transfected by electroporation using nucleofection technique according to the manufacturer’s instruction and optimization techniques. Briefly, the cells were transferred to 24-well plate delivered with Nucleofection Kit (Amaxa, SF Cell Line 4D-Nucleofector X Kit L, cat. No. V4XC-2024, Lonza, Basel, Switzerland) (Amaxa, SE Cell Line 4D-Nucleofector X Kit L, cat. No. V4XC-1024, Lonza, Basel, Switzerland). The wells containing cells were supplemented with Nucleofector solutions and MMP-9 siRNA (cat. No. sc-29400, Santa Cruz, Dallas, TX, USA). The plate was covered with Dipping Electrode Array, put into Nucleofector and electroporated using dedicated impulse program. The programs were dependent on the type of the cell and defined by manufacturer (MCF-7: EN-130, MDA-MB-231: CH-125, MCF-10A: T-024). As a negative control the control siRNA (cat. No. sc-37007, Santa Cruz, Dallas, TX, USA) was used. Semi-quantitative analysis of post-translational MMP-9 expression was analyzed by Western blot according to the methodology described below.

### 4.3. TCGA, GTEx and CPTAC Analysis

To assess the expression of MMP-9 and KRT19 (CK19) and their clinical relevance in breast cancer we analyzed the TCGA, GTEx, and CPTAC data. The TCGA and GTEx mRNA expression data were retrieved via Xena Functional Genomics Explorer [36] and CPTAC protein data were obtained from the UALCAN database [37]. The cutpoints for the analysis were calculated using Evaluate Cutpoint R package [38]. The statistical analysis for RNA-seq data was performed in GraphPad Prism 9 and CPTAC statistical analysis was taken directly from the database.

### 4.4. MTT Assay

Cells were seeded in 24-well plates, cultured for 24 h, and treated with CP for another 24 h. The increasing drug doses used in the studies were: 0.1 mM, 0.5 mM, 1 mM, 2 mM, 2.5 mM, 5 mM, 8 mM, 10 mM. A stock solution of CP was prepared by dissolving the powder in 100% DMSO to a final 182.7 mM concentration. The dilutions were prepared in a complete growth medium immediately before use. The MTT salt was dissolved (50 mg/mL) in phosphate-buffered saline (PBS, cat. No. 21-040-CV, Corning, New York, NY, USA), diluted 1:9 in phenol red-free DMEM (Corning, New York, NY, USA) right before the experiment and 500 μL were added to each well for 3 h (incubator). The resulting purple formazan crystals were dissolved with dimethyl sulfoxide solution (DMSO, cat. No. 210520182, CHEMPUR, Piekary Śląskie, Poland). The results of the MTT assay have been documented using BioTEK 800Ts fluorescence microplate reader (BioTEK, Bad Friedrichshall, Germany) at 570 nm wavelength.

### 4.5. Cell Death Analysis

The cell death (apoptosis, necrosis) was analyzed by flow cytometry using Guava^®^ easyCyte™ 6HT-2L system and InCyte Software (version 3.3, Merck KGaA, Darmstadt, Germany). The cells were seeded in the 12-well plates. The next day cells were treated with a 5 mM concentration of the CP and incubated for 24 h. Then, the cells were detached. The cell suspension was centrifuged at 300 RCF for 5 min at room temperature (RT). The pellet was resuspended in 200 µL of Annexin Binding Buffer (ABB, cat. No. V13246, Thermo Fisher Scientific, Waltham, MA, USA). The cells were stained using propidium iodide solution (cat. No. P1304MP, Thermo Fisher Scientific, Waltham, MA, USA) and Annexin V Alexa Fluor 488 (cat. No. A13201, Thermo Fisher Scientific, Waltham, MA, USA) following the manufacturer’s instructions (15 min, RT, dark). Next, the cell suspension was transferred to a 96-well plate (cat. No. 0030601106, Eppendorf, Hamburg, Germany) for the cytometric analysis.

### 4.6. Cell Cycle Analysis

The cell cycle was analyzed using flow cytometry. MCF-7 and MDA-MB-231 cells were cultured into 12-well plates. After reaching 70% confluence, the cells were treated with CP and incubated for 24 h. Then, the cells were detached, centrifuged (300 RCF, 5 min, RT), and fixed in 1 mL 70% ethanol (cat. No. PA-06-396480427-25L, Pol-Aura, Różnowo, Poland) for 24 h at −20 °C. The fixed cells were centrifuged (500 RCF, 7 min, RT) to remove ethanol. The pellet was washed with PBS. Next, the cells were suspended in 200 μL FxCycle PI/RNase Staining Solution (cat. No. F10797, Thermo Fisher Scientific, Waltham, MA, USA) and incubated according to the Manufacturer’s instruction (30 min, RT, dark). The cells were placed in a 96-well plate and analyzed using Guava^®^ easyCyte™ 6HT-2L system and InCyte Software (version 3.3, Merck KGaA, Darmstadt, Germany).

### 4.7. Immunoblotting Technique

Western blot was used to assess the efficiency of transfection with MMP-9 siRNA and to measure CK19, vimentin and N-cadherin protein level. The cells used in the experiment were transfected using the protocol described above and seeded in T-24 culture bottles. After 24 h, the cells were treated with the CP solution and incubated for another 24 h. Control cells were cultured under the same conditions but without CP addition. The lysis mixture contained RIPA buffer (cat. No. 20-188, Merck KGaA, Merck KGaA, Darmstadt, Germany) and Halt protease inhibitor cocktail (cat. No. 78438, Thermo Scientific, Waltham, MA, USA). The lysis was carried out for 10 min at 4 °C. Next, the samples were centrifuged (10.000 RCF, 10 min, 4 °C). The supernatant was used for the quantitative determination of the concentration of proteins in samples by the Pierce™ BCA Protein Assay Kit (cat. No. 23225, Thermo Fisher Scientific, Waltham, MA, USA) according to the manufacturer’s recommendations. The quantitative measurement of protein resulted in a standard curve, which allowed to calculate the volume of each sample and standardize the amount of protein taken for analysis. The samples were prepared by mixing appropriate volumes of cell lysates, NuPage Sample Reducing Agent (cat. No. NP0009, Thermo Fisher Scientific, Waltham, MA, USA), Tris-Glycine SDS Sample Buffer (cat. No. LC2676, Thermo Fisher Scientific, Waltham, MA, USA), and distilled water. The heat denaturation of the samples was carried out at 85 °C for 3 min. The denatured samples were placed in the polyacrylamide gel (Novex™ WedgeWell™ 10–20% Tris-Glycine Gel 1.0 × 10 well, XP10200BOX, Thermo Fisher Scientific, Waltham, MA, USA). PageRuler™ Prestained Protein Ladder, 10 to 180 kDa (cat. No. 26616, Thermo Fisher Scientific, Waltham, MA, USA) was used to assess the protein molecular weight. The electrophoretic separation was carried out for 30 min at 225 V in Tris-Glycine SDS Running Buffer (cat. No. LC2675-5, Thermo Fisher Scientific, Waltham, MA, USA). The next step was dry transfer onto a nitrocellulose membrane using the iBlot Dry Western Blotting System (7 min, 10 V). Transfer efficiency was checked by staining the nitrocellulose membrane with Ponceau S solution (cat. No. P7170, Merck KGaA, Merck KGaA, Darmstadt, Germany). The membranes were incubated in an enhancer solution for 20 min. Primary (anti- MMP-9: cat. No. sc-21733, Santa Cruz, Dallas, TX, USA; anti-CK9: cat. No. MA5-12663, Thermo Fisher Scientific, Waltham, MA, USA; anti-vimentin: cat. No. MA5-16409, Thermo Fisher Scientific, Waltham, MA, USA; anti-N-cadherin: cat. No. 33-3900, Thermo Fisher Scientific, Waltham, MA, USA, and secondary antibodies HRP (cat. No. 31460, cat. No. 31430, Thermo Fisher Scientific, Waltham, MA, USA) were diluted in the iBind Flex Solution Kit (cat. No. SLF2020, Thermo Fisher Scientific, Waltham, MA, USA) according to the manufacturer’s instructions. The membranes were applied onto iBind Flex Cards (cat. No. SLF2010, Thermo Fisher Scientific, Waltham, MA, USA) and placed in the iBind Automated Western System (Thermo Fisher Scientific, Waltham, MA, USA) for 24 h at 4 °C. Finally, the results were visualized using enhanced chemiluminescence (ECL, cat. No. 32109, Thermo Fisher Scientific, Waltham, MA, USA) detection system and documented by the ChemiDoc MP Imaging System (Bio-Rad, Hercules, CA, USA ). The intensity of the bands was measured in ImageLab (Bio-Rad, Hercules, CA, USA).

### 4.8. Immunofluorescent Staining

The cells were seeded into 12-well plates and grown on sterile ⌀18 mm glass coverslips. After 24 h, the cells were treated by the 5 mM CP concentration for another 24 h (incubator). Coverslips were fixed in 4% paraformaldehyde (PFA, cat. No. 28906, Thermo Fisher Scientific, Waltham, MA, USA; 20 min, RT). All further steps were preceded by washing the cells three times with PBS (5 min, RT). The fixed cells were permeabilized with the 0.25% Triton X-100 in PBS (10 min, RT). To block non-specific interactions 5% BSA (cat. No. 9048-46-8, Merck KGaA, Merck KGaA, Darmstadt, Germany) in PBS was used (20 min, RT). The next step was incubation with CK19, vimentin and N-cadherin primary antibodies (1:100 in BSA, 1 h, RT). Then, the cells were incubated with secondary antibodies (Alexa Fluor 594, Thermo Fisher Scientific, Waltham, MA, USA, 1:200 in PBS, 1 h, RT, dark). Alexa Fluor 488 phalloidin (cat. No. A12379,) was used for the labeling of F-actin (1:20 in PBS, 20 min, RT, dark). Cell nuclei were stained with DAPI (1:100 in distilled water, 10 min, RT, dark). The coverslips were mounted with Aqua Poly/Mount (cat. No. 18606-5, Polysciences) and photographed using the C1 confocal microscope (Nikon, Tokyo, Japan) and 100× oil immersion objective. The images were collected at the brightest signals of protein using Nikon EZ-C1 software (Nikon, Tokyo, Japan). Laser power, gains, and pixel dwell time were kept constant for all repeats for each protein.

To quantitatively confirm the fluorescence level of the selected proteins, analogous labeling steps were performed on the suspension of transfected and non-transfected untreated and treated MCF-7 and MDA-MB-231 cells. The staining results were measured by the microcapillary flow cytometry system and analyzed using FlowJo software (version 10.07, BD Biosciences, Frankin Lakes, NJ, USA).

### 4.9. Transwell Cell Migration and Invasion Assays

The migration and invasion of the non-tumorigenic cells and mammary gland cancer cell lines were assessed using cell culture inserts (cat. No. 140620, Thermo Fisher Scientific, Waltham, MA, USA). The inserts were put into a 24-well plate. Control and treated cells were transfected and non-transfected cells were plated on inserts at a density of 105 cells per well in a culture medium supplemented with 1% FBS. Under the inserts, a medium with 20% FBS was added to attract the cells to pass through the insert pores. For the invasion assay Matrigel (cat. No. 354277, Corning, New York, NY, USA) was prepared by diluting the stock solution in the 1:4 ratio in PBS, poured onto the inner surface of the insert, and polymerized for 24 h. The control and treated cells were plated on the surface of the Matrigel at a density analogous to the cell migration assay. The experimental assessment of the invasive potential of the cells was carried out for 24 h in standard culture conditions (95% humidity, 5% CO_2_, and 37 °C).

In both experiments, transwells were fixed with a 3.7% paraformaldehyde solution. The cells were permeabilized using the 100% ice-cold methanol (621990426, Avantor, Toruń, Poland) solution (20 min, RT). The next step was staining with the 0.4% crystal violet (cat. No. C0775, Thermo Fisher Scientific, Waltham, MA, USA) solution (15 min, RT). Noninvasive/nonmigrating cells were removed from the inner insert surfaces using the cotton swabs. The outer surface of the insert membrane was photographed using a light microscope with a CCD digital camera (DS-5Mc-U1). The percentage values of invasive and migrating cells were calculated using ImageJ software (NIH).

### 4.10. Clonogenic Assay

Transfected and non-transfected MCF-7 and MDA-MB-231 cells were treated with 5 mM concentration of the CP for 24 h or left untreated in the case of control cells. Then, the cells were seeded into Petri dishes at a density of 1 × 103 cells and cultured for 14 days in standard conditions (37 °C and 5% CO_2_). The resulting colonies were fixed with 4% paraformaldehyde in PBS (20 min, RT), stained using a 0.4% crystal violet in PBS (15 min, RT), and washed with PBS. The results were documented using ChemiDoc MP Imaging System. The number of colonies was measured by ImageJ software (NIH).

### 4.11. Wound Healing

The transfected and non-transfected cells of both lines were seeded and grown in 6-well plates until reaching 100% confluency in standard cell culture conditions. The 100 μL sterile pipette tips were used to scratch the surface in the center of each well. The cells were treated with 5 mM CP or left untreated in the case of control cells. Experiment was performed in the inverted motorized microscope (Zeiss) equipped with an incubation system for live-cell imaging (Pecon) containing EC Plan-Neofluar 10×/0.30 Ph1 air objective, Axiocam 503 mono camera, and ZEN 2 software. The wound surface and other parameters were measured using ImageJ software.

### 4.12. Statistical Analysis

All procedures were performed in at least three independent replicates. In the case of significantly different results, they were discarded, and the number of repetitions increased accordingly. The obtained results were analyzed with the GraphPadPrism 9 Software. The significance was assumed at *p* > 0.05. The data were compared using the Kruskal–Wallis test with the exception of the apoptosis data (Kruskal–Wallis test with Dunn’s multiple comparisons test), the MTT assay (Wilcoxon test), and the cell cycle data (two-way ANOVA).

## 5. Conclusions

In conclusion, our research indicates the cytotoxic and antimetastatic nature of CP in cells with reduced MMP-9 levels in MCF-7 and MDA-MB-231 breast cancer cell lines. In addition, we observed a positive correlation between MMP-9 and CK19. Breast cancer cell lines used in the study are characterized by the high expression of both proteins, and after downregulation of MMP-9 level, the expression of CK19 also decreased.

## Figures and Tables

**Figure 1 ijms-22-12783-f001:**
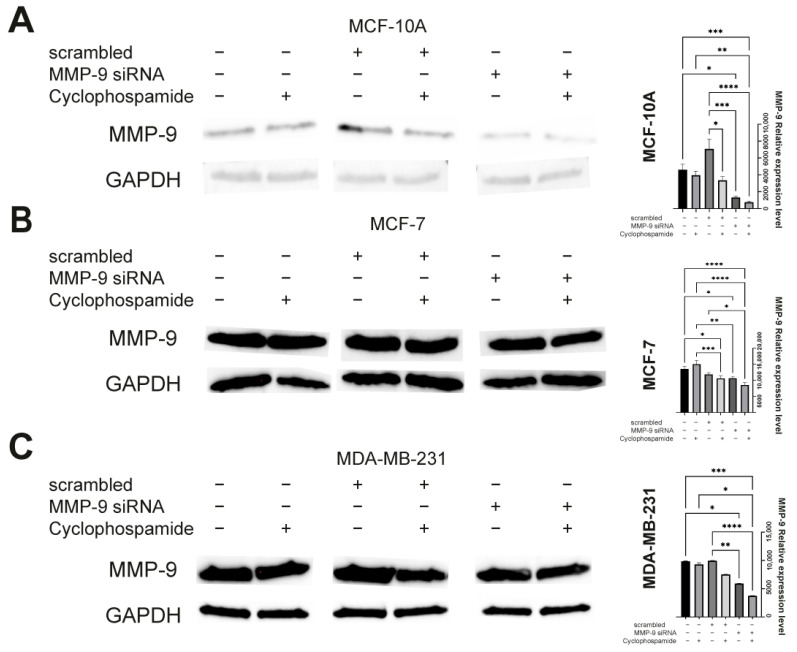
The Western blot verification of MMP-9 silencing. (**A**) MCF-10A, (**B**) MCF-7, and (**C**) MDA-MB-23 cells non-transfected, transfected with control siRNA (scrambled) and transfected with MMP-9 siRNA treated with cyclophosphamide at 5 mM concentration. Statistically significant differences were marked with ‘*’ *p* ≤ 0.05, ‘**’ *p* ≤ 0.01, ‘***’ *p* ≤ 0.001, and ‘****’ *p* ≤ 0.0001 (Kruskal-Wallis test).

**Figure 2 ijms-22-12783-f002:**
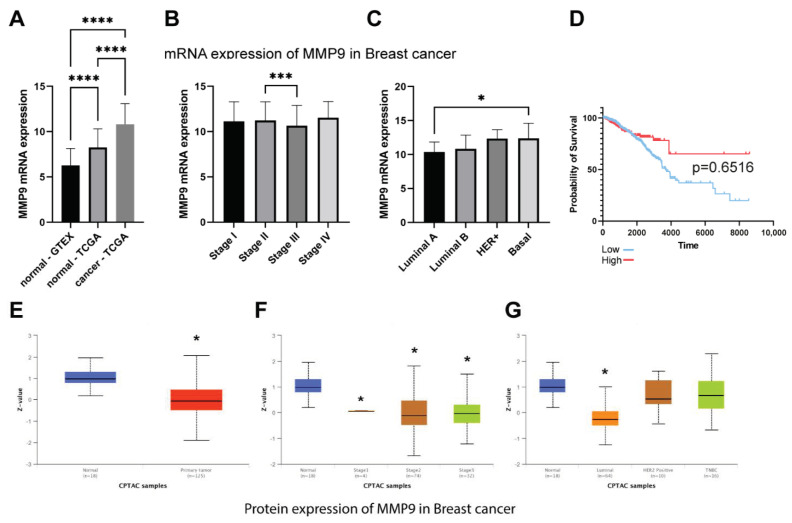
The expression of MMP-9 mRNA in GTEx and TCGA breast cancer cohort. (**A**) MMP-9 expression in normal and tumor tissue, (**B**) MMP-9 expression in different tumor stages, (**C**) The comparison of MMP-9 expression among PAM50 subtypes. (**D**) Overall survival of TCGA breast cancer cohort. There were no significant differences for patients with low and high MMP-9 mRNA expression. The survival curves were compared using Log-rank (Mantel-Cox) test. *p* < 0.05 was considered significant. The analysis of MMP-9 proteins in breast cancer patients (CPTAC cohort). (**E**) MMP-9 expression in normal and tumor tissue, (**F**) MMP-9 expression in different tumor stages, (**G**) The comparison of MMP-9 expression among PAM50 subtypes. The asterisks point to statistically significant differences (Kruskal-Wallis test with Dunn’s multiple comparison test). ‘*’ *p* ≤ 0.05, ‘***’ *p* ≤ 0.001, and ‘****’ *p* ≤ 0.0001.

**Figure 3 ijms-22-12783-f003:**
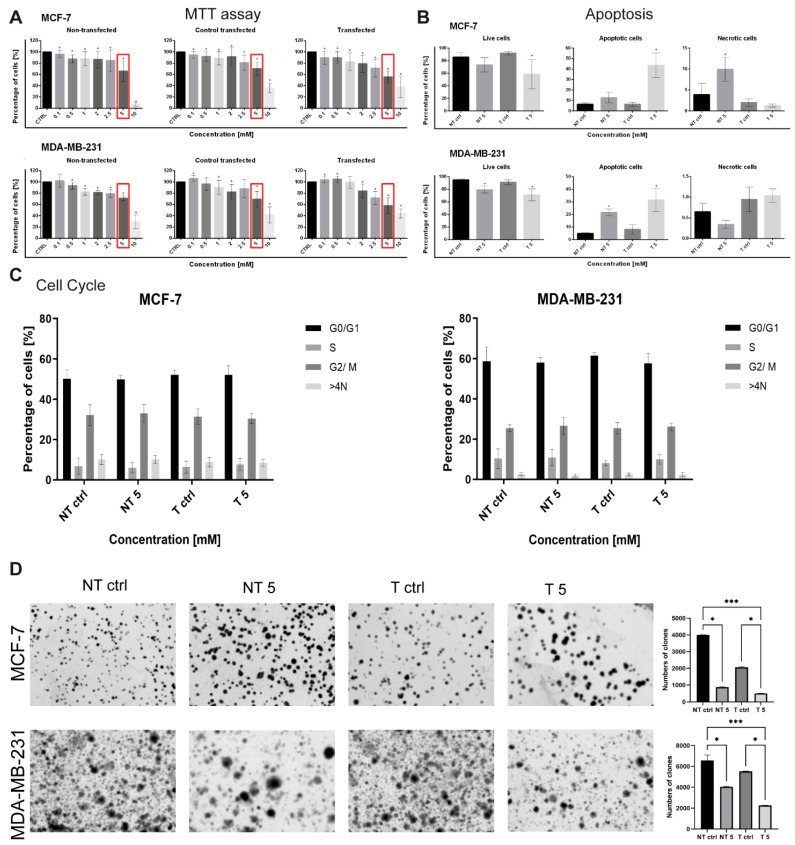
(**A**) The cell viability analysis was based on the MTT assay. The analysis included non-transfected, transfected with control siRNA, and transfected with MMP-9 siRNA cells treated with cyclophosphamide at 0.1–10 mM concentrations for 24 h. The data represent mean values ± SD of 4 independent experiments (*n* = 4). Statistically significant differences were marked with ‘*’ (*p* < 0.05; Wilcoxon test). (**B**) The cell death analysis was based on AnnexinV/PI staining. NT—non-transfected cells, T—cells transfected with MMP-9 siRNA, NT5 and T5 non-transfected and transfected cells treated with 5 mM cyclophosphamide for 24 h. Data represent the mean ± SD obtained from 4 independent replicates (*n* = 4). Statistically significant differences are marked with ‘*’ (*p* < 0.05; Kruskal–Wallis test with Dunn’s multiple comparisons test). (**C**) The cell cycle analysis of MCF-7 and MDA-MB-231cells. NT—non-transfected cells, T—cells transfected with MMP-9 siRNA, NT5 and T5 non-transfected and transfected cells treated with 5 mM cyclophosphamide for 24 h. Data represent the mean ± SD obtained from 4 independent replicates (*n* = 4, statistical Two-way ANOVA test). (**D**) The effect of MMP-9 downregulation on colonies formed of breast cancer cell lines. NT—non-transfected cells, T—cells transfected with MMP-9 siRNA, NT5 and T5 non-transfected and transfected cells treated with 5 mM cyclophosphamide for 24 h. Representative image of clonogenic assay and the number of colonies formed. Statistically significant differences were marked with ‘*’ *p* ≤ 0.05, ‘***’ *p* ≤ 0.001 (Kruskal-Wallis test).

**Figure 4 ijms-22-12783-f004:**
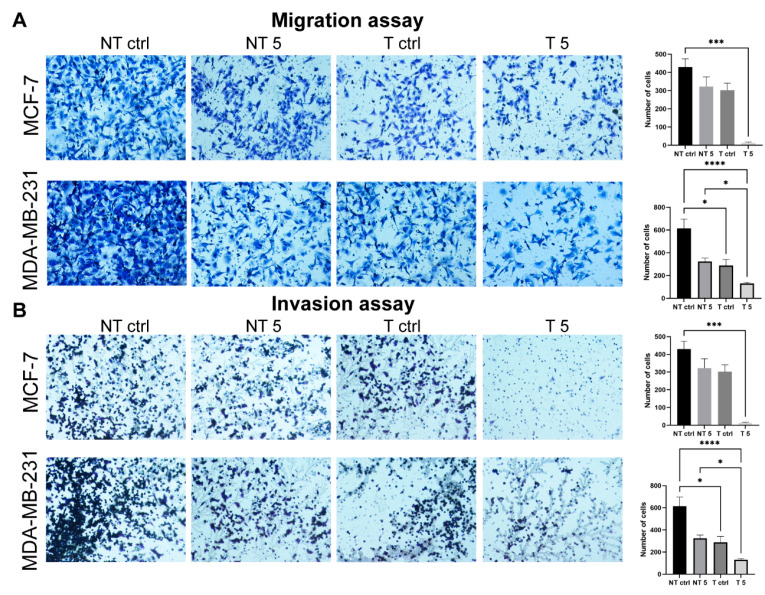
The effect of MMP-9 downregulation on metastasis potential of MCF-7 and MDA-MB-231 breast cancer cell lines. (**A**) migration and (**B**) invasion assay. NT—non-transfected cells, T—cells transfected with MMP-9 siRNA, NT5 and T5 non-transfected and transfected cells treated with 5 mM cyclophosphamide for 24 h. Magnification 400× (**A**) Representative image of transwell migration assay and number of cells with high migratory potential. Data represents the mean ± SD obtained from 6 independent replicates (*n* = 6). Statistically significant differences were marked with ‘*’ *p* ≤ 0.05, ***’ *p* ≤ 0.001, and ‘****’ *p* ≤ 0.0001 (Kruskal-Wallis test). (**B**) Representative image of invasion assay and number of cells with high invesion potential. Data represents the mean ± SD obtained from 6 independent replicates (*n* = 6). Statistically significant differences were marked with ‘*’ *p* ≤ 0.05, ‘***’ *p* ≤ 0.001, and ‘****’ *p* ≤ 0.0001 (Kruskal-Wallis test).

**Figure 5 ijms-22-12783-f005:**
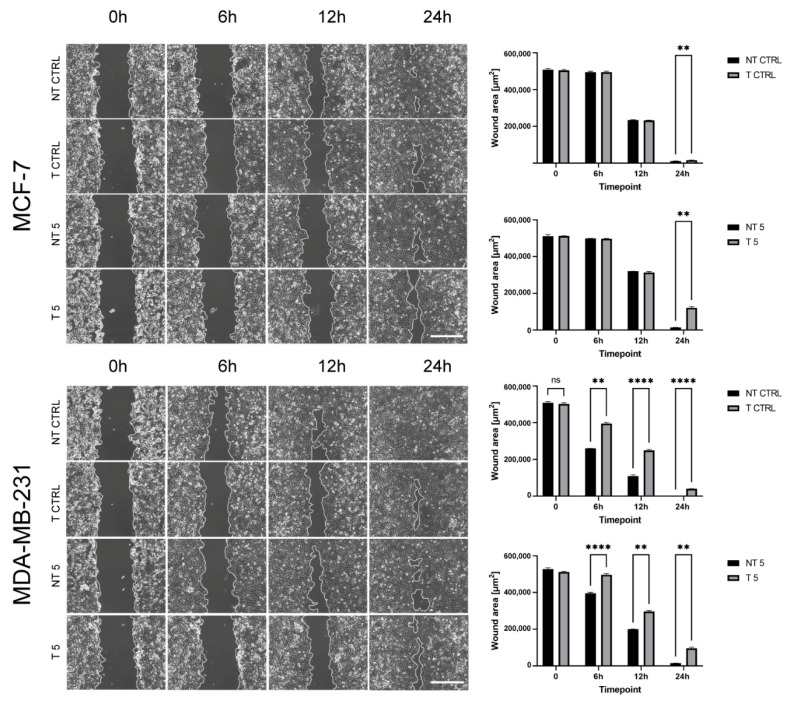
The effect of MMP-9 downregulation on metastasis potential of breast cancer cell lines—Wound healing assay. NT—non-transfected cells, T—cells transfected with MMP-9 siRNA, NT5 and T5 non-transfected and transfected cells treated with 5 mM cyclophosphamide for 24 h. Representative image of Wound healing assay (Bar = 50 μm) and wound area in timepoints. Statistically significant differences were marked with ‘**’ *p* ≤ 0.01, and ‘****’ *p* ≤ 0.0001 (Kruskal-Wallis test).

**Figure 6 ijms-22-12783-f006:**
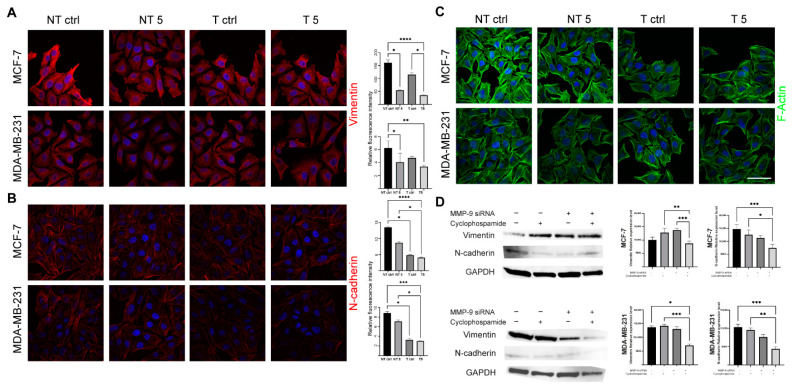
The effect of MMP-9 downregulation on vimentin, N-cadherin and F-actin level. NT—non-transfected cells, T—cells transfected with MMP-9 siRNA, NT5 and T5 non-transfected and transfected cells treated with 5 mM cyclophosphamide for 24 h. (**A**) Immunofluorescent imaging of vimentin (red) and nuclei (blue), and relative fluorescence intensity of vimentin. (**B**) Immunofluorescent imaging of N-cadherin (red) and nuclei (blue), and relative fluorescence intensity of N-cadherin. (**C**), Immunofluorescent imaging of F-actin (green), and nuclei (blue), Bar = 50 μm. (**D**) The Western blot analysis of vimentin and N-cadherin and densitometric relative expression level of the Western blot experiments. Statistically significant differences were marked with ‘*’ *p* ≤ 0.05, ‘**’ *p* ≤ 0.01, ‘***’ *p* ≤ 0.001, and ‘****’ *p* ≤ 0.0001 (Kruskal-Wallis test).

**Figure 7 ijms-22-12783-f007:**
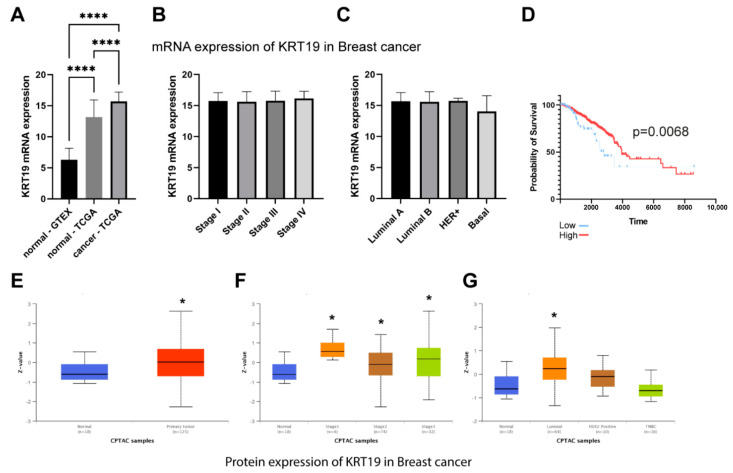
The expression of KRT19 (CK19) mRNA in GTEx and TCGA breast cancer cohort. (**A**) KRT19 expression in normal and tumor tissue, (**B**) KRT19 expression in different tumor stages, (**C**) The comparison of KRT19 expression among PAM50 subtypes. (**D**) Overall survival of TCGA breast cancer cohort. Lower KRT19 (CK19) mRNA expression is associated with better outcomes for breast cancer patients. The survival curves were compared using Log-rank (Mantel-Cox) test. *p* < 0.05 was considered significant. The analysis of KRT19 proteins in breast cancer patients (CPTAC cohort). (**E**) KRT19 expression in normal and tumor tissue, (**F**) KRT19 expression in different tumor stages, (**G**) The comparison of KRT19 expression among PAM50 subtypes. The asterisks point to statistically significant differences comparing to the normal group (Kruskal-Wallis test with Dunn’s multiple comparison test). ‘*’ *p* ≤ 0.05, ‘****’ *p* ≤ 0.0001.

**Figure 8 ijms-22-12783-f008:**
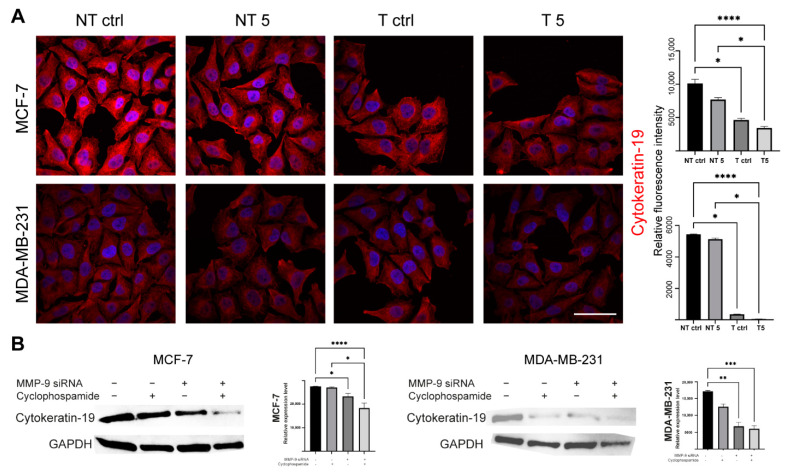
The effect of MMP-9 downregulation on Cytokeratin 19 (CK19) level. NT—non-transfected cells, T—cells transfected with MMP-9 siRNA, NT5 and T5 non-transfected and transfected cells treated with 5 mM cyclophosphamide for 24 h. (**A**) Immunofluorescent imaging of Cytokeratin 19 (red), and nuclei (blue), and cytometric relative fluorescence intensity of Cytokeratin 19, Bar = 50 μm. (**B**) The Western blot analysis of Cytokeratin 19 and densitometric relative measurement of the Western blot experiments. Statistically significant differences were marked with ‘*’ *p* ≤ 0.05, ‘**’ *p* ≤ 0.01, ‘***’ *p* ≤ 0.001, and ‘****’ *p* ≤ 0.0001 (Kruskal-Wallis test).

## Data Availability

The datasets used during the present study are available from the corresponding author upon reasonable request.

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
