# Peer review of "Downregulation of MMP-9 Enhances the Anti-Migratory Effect of Cyclophosphamide in MDA-MB-231 and MCF-7 Breast Cancer Cell Lines"

_ijms, 2021, doi:10.3390/ijms222312783_

Round 1
Reviewer 1 Report
Downregulation of MMP-9 enhances the anti-migratory effect 2 of cyclophosphamide in MDA-MB-231 and MCF-7 breast cancer cell lines by Izdebska et al is interesting. However, there are major issues with the experimental designs.
Comment 1) MMP-9 siRNA and cyclophosphamide effect are inconclusive, it needs to run on the same gel.
Comment 2) Thre are no in vivo evidence to support in vitro data
Author Response
Thank you very much for all your valuable comments. It undoubtedly helped us to improve our manuscript. All of the suggestions were carefully considered by our research team.
Point 1: MMP-9 siRNA and cyclophosphamide effect are inconclusive; it needs to run on the same gel.
A: Every protein was run on a single gel. The wells were cut for the publication purpose to put the bands in line. We provide full-size MMP-9 original gels that have also been sent to MDPI Editors as it is required by submission rules.
Point 2: There are no in vivo evidence to support in vitro data
A: Unfortunately, we have no technical possibilities to conduct the in vivo experiments. However, we believe that we have submitted concise and convincing evidence of the influence of MMP-9 downregulation on different breast cancer cell lines. We are naturally aware that with in vivo studies our paper would be more conclusive, but we do not feel that the lack of such an experiment makes our conclusions misleading or overinterpreted. Thus, the presented data are valuable and worthy to share with the scientific community, which may encourage the other team to perform studies providing more information about the mechanistic details of oncogenic properties of MMP-9 in breast cancer.

Reviewer 2 Report
1. Why did cyclophosphamide use this concentration of 5 mM in Figure 1?
2. What is the reason why breast cancer cells did not die after transfection of MMP9 siRNA and the cell clone formation was influenced by MMP9 siRNA in Figure 3?
3. Apoptosis scatter plot and cell cycle peak plot were not displayed.
Author Response
Thank you very much for all your valuable comments. It undoubtedly helped us to improve our manuscript. All of the suggestions were carefully considered by our research team.
Point 1: Why did cyclophosphamide use this concentration of 5 mM in Figure 1?
A: The 5 mM dose has been chosen based on the results of experiments presented in Figure 3. We understand that such an order can make some confusion. We decided on such a layout because we wanted to show a direct comparison between non-transfected, and scramble siRNA and MMP-9 siRNA-transfected cells. We intended to put the data that confer to cell proliferation altogether rather than split the “initial” results from downregulation experiments and create additional graphs which made the figures less clear.
Point 2: What is the reason why breast cancer cells did not die after transfection of MMP9 siRNA and the cell clone formation was influenced by MMP9 siRNA in Figure 3?
A: MMP-9 is not a housekeeping gene so there is no reason that control cells should undergo apoptosis after the silencing. Simultaneously, the decreased ability to colony formation is a result of lower proliferation rates rather than the effect of cytotoxicity. The increased doubling time resulted in fewer colonies. The decrease in proliferation rate does not affect the shape of the cell cycle plot, because cell division is normal, there is no block in any particular cell cycle phase.
Point 3: Apoptosis scatter plot and cell cycle peak plot were not displayed.
A: Thank you for your advice. We provided the representative plots in Supplementary Figure S1.

Reviewer 3 Report
Good publication on the role of MMP9 but requiring some additional data. Authors are advised to add and conduct experiments with one line of breast cancer with different MMR9 expression.
Author Response
Point 1: Authors are advised to add and conduct experiments with one line of breast cancer with different MMR9 expression.
A: Thank you very much for your valuable comment. The suggestion was carefully considered by our research team. Unfortunately, we are unable to provide any additional analysis due to our funding source requirements and limitations. However, we believe that our study shows significant data related to the role of MMP-9 in breast cancer and is worthy to share with the scientific community. All of the experiments were carried out on the two breast cancer cell lines of different invasiveness and MMP-9 expression (MDA-MB-231 with high invasiveness and MMP-9 expression and MCF-7 with low invasiveness and MMP-9 expression). Additionally, the results were compared with the control cell line MCF-10A. We are planning to investigate the precise mechanisms of MMP-9 action in cancer cells in the nearest future.

Round 2
Reviewer 1 Report
The manuscript quality improved
Reviewer 2 Report
There are some other questions in this article.